# Neuroprotection and Neuroregeneration Strategies Using the rNAION Model: Theory, Histology, Problems, Results and Analytical Approaches

**DOI:** 10.3390/ijms232415604

**Published:** 2022-12-09

**Authors:** Steven L. Bernstein, Yan Guo, Zara Mehrabian, Neil R. Miller

**Affiliations:** 1Department of Ophthalmology and Visual Sciences, University of Maryland at Baltimore School of Medicine, 10 S. Pine St., Baltimore, MD 21201, USA; 2Department of Anatomy and Neurobiology, University of Maryland at Baltimore School of Medicine, 10 S. Pine St., Baltimore, MD 21201, USA; 3Wilmer Eye Institute, Johns Hopkins University School of Medicine, 600 N. Wolfe St., Baltimore, MD 21205, USA

**Keywords:** nonarteritic anterior ischemic optic neuropathy, retinal ganglion cells, optic nerve, neuroprotection, rodents, gene expression, models

## Abstract

Nonarteritic anterior ischemic optic neuropathy (NAION) is the most common cause of sudden optic nerve (ON)-related vision loss in humans. Study of this disease has been limited by the lack of available tissue and difficulties in evaluating both treatments and the window of effectiveness after symptom onset. The rodent nonarteritic anterior ischemic optic neuropathy model (rNAION) closely resembles clinical NAION in its pathophysiological changes and physiological responses. The rNAION model enables analysis of the specific responses to sudden ischemic axonopathy and effectiveness of potential treatments. However, there are anatomic and genetic differences between human and rodent ON, and the inducing factors for the disease and the model are different. These variables can result in marked differences in lesion development between the two species, as well as in the possible responses to various treatments. These caveats are discussed in the current article, as well as some of the species-associated differences that may be related to ischemic lesion severity and responses.

## 1. Introduction

Nonarteritic anterior ischemic optic neuropathy (NAION) is an ischemic lesion in the anterior portion of the optic nerve, which results in sudden vision loss, visible edema of the optic nerve head (the optic disc), and ultimately retinal ganglion cell (RGC) neuron death and loss of RGC axons by Wallerian degeneration (progressive axon degeneration in both directions from the site of the lesion). Most clinical NAION cases are believed to be due to a post-ischemic compartment syndrome associated with vascular (capillary) dysregulation at the site of the initial lesion [1]. However, a number of associated factors potentially contribute to this disease, and the relative degree of severity of each case. These include the size of the optic nerve as it enters the eye (morphology), vascular dysregulation due to underlying systemic disorders such as hypertension and diabetes, familial genetics and mitochondrial mutations, and possible other spontaneous pathologic processes. These include such things as microemboli to the optic nerve head, elevated central venous pressure, and differential cerebrospinal fluid pressure gradients [2,3,4]. Importantly, no currently available drug has been shown to be consistently effective in reducing the damage or in ameliorating the visual debilitation that occurs in NAION. One reason for this lack of progress in clinical treatment is a dearth of knowledge of the pathophysiological mechanisms responsible for permanent visual loss after NAION. Another reason is the variable time from onset of NAION symptoms to clinical recognition and possible treatment. Finally, it is clear that some patients have some degree of optic disc swelling, for an unknown amount of time, before clinical symptoms develop. Thus, we largely lack an understanding of the processes and time window opportunities that may be available for different treatment approaches. The rodent NAION (rNAION) and primate NAION (pNAION) models are now available for intensive analysis of pathophysiological mechanisms associated with clinical NAION, evaluation of potential treatments, and to identify the time windows of opportunity for treating this disorder [5,6,7]. 

### The rNAION Model: Histology and Comparison with Human Disease

The optic nerve head (ONH) has a vasculature far more complex than the distal optic nerve, with circulatory contributions from the retina, choroid and optic nerve-pial vessels. Thus there are possibilities for dysregulation from multiple components (see Figure 1). 

The ONH is bounded within the thick-walled sclera and in primates, the individual axon bundles are enclosed between collagenous columns. Thus there is little room for compensatory tissue expansion in the setting of edema. This problem is compounded in individuals with only a small opening in the sclera where the optic nerve emerges to form the optic nerve (the ‘disk-at-risk’) [3]. 

In prototypical human NAION, vascular dysregulation anywhere in the optic nerve head can result in some degree of interstitial (extravascular) or intracellular edema. This causes expansion of the fascicles containing the ON axons until resistance is met by either the tissue surrounding the fascicles or the optic nerve sheath. At that point, extracellular pressure increases by hydraulic action until the resistance is sufficient to close off capillary circulation, resulting in a compartment syndrome with tissue ischemia. Human ON axons are thickly fasciculated: that is, individual groups of axons (fascicles) are separated by relatively dense collagenous partitions within the nerve. Thus, compartment syndromes can occur in only a portion of the human ON, affecting individual fascicles. 

The farther away in evolution an animal model is from humans, the more retinal and optic nerve dissimilarities may occur that can confound both the interpretations of pathology and possible treatments. Thus, animal models of NAION are just that: Models that are representative of various components of the disease and potentially useful for analysis of disease mechanisms and treatments, but that must always be considered with a number of caveats, including reliance on individual pathways to a greater or lesser degree. Although rats and humans share ~99% of their overall genes [8], different species can express individual genes that may be relevant to a specific neuroprotective mechanism at different levels. For example, L-type calcium channel blockers are highly effective neuroprotectants in models of rodent central nervous system (CNS) ischemia [9] but relatively ineffective in humans [10]. This may be partly because rats express L-type calcium channels in ON tissue at levels ~5-fold higher than in humans or other old world primates [11], suggesting that rats rely on these channels to a greater degree than primates. 

The initial NAION model was defined in rats and called rodent anterior ischemic optic neuropathy (rAION) or rodent NAION (rNAION). Both terms are used in the literature. The intraocular optic disk diameter in the rat examined through a plano-convex contact lens is 500 µm (Figure 1B, arrows), but narrows within the sclera to ~250 µm (Figure 1B). A 500 µm laser spot thus irradiates both the optic nerve and the peripapillary retina and its supporting vessels. While the collagen sheaths in the human ON surrounding the axon fascicles are quite dense, the sheaths surrounding rodent ON axons are extremely thin. The laser/rose Bengal-induced vascular decompensation causes focal serum leakage in the ONH (Figure 1C), resulting in intraneural edema limited by the sclera and ON sheath (Figure 1C). This gross expansion is easily seen in vivo using spectral-domain optical coherence tomography (SD-OCT), where it can be measured at the anterior portion of the ON, including its intraretinal portion (compare Figure 2F, control rat ONH, with Figure 2H; post-induction). ONH capillaries are likely compromised by a rise in overall intraneural pressure, rather than from compromise of individual fascicles. Axon loss in rNAION thus is likely a function of the total size of the nerve, rather than associated with individual regions, as occurs in humans and in the nonhuman primate (NHP) NAION model (pNAION) [7,12]. The laser spot must be precisely focused; if laser irradiation extends over the surrounding peripapillary retina, this can result in decompensation of retinal vasculature, causing both ON and retinal ischemia. Including animals with mixed ischemic lesions in the evaluation of rNAION can confound individual analyses and confuse subsequent treatment approaches. In particular, as retinal responses to neuroprotective agents can be vastly different than those of the ON, it can be difficult to separate treatment responses to the retinal ischemic and optic nerve head-axon ischemic components. A ganzfeld electroretinogram can be used to distinguish animals with mixed lesions from those with pure ON ischemia. 

Mice ONs have fewer axons than rat ONs (mice: ~50,000 vs. rat: ~100,000), and a correspondingly smaller ONH, which means that a 300µm laser spot size at 50 mW is required for mouse induction, rather than the 500 µm spot used for rats. The same 50 mW of radiant power is distributed over a smaller area (mouse ON circumference is only 36% the size of the rat ON). There is, therefore, correspondingly more power per unit area, and a potential for greater capillary decompensation. Additionally, ON ischemic induction that results in large percentages (>75%) of RGC loss can be associated with amacrine retinal neuron loss [13]. This suggests that conditions required to generate severe ONH edema also can induce a state of overlapping retinal ischemia. Following mouse rNAION induction, great care is needed to identify animals with pure ON ischemic lesions, so that animals with mixed retinal and ON ischemia can be eliminated from further analyses. 

Some of the advantages of the rNAION model, then, are balanced by the differences from human NAION as well as pNAION in ON morphology. Identifying additional species whose ONH and ON structure more closely resemble that of primates may help reduce some of the drawbacks associated with the rodent model, but the rNAION model, with its ease of induction, the relative inexpense of the animals and of animal upkeep, and the availability of complete genomic information for both rats and mice make this model an invaluable tool to use in ischemic optic neuropathy research. 

## 2. Results

### 2.1. Analytical Approaches and Correlation

Although the initiating events that result in ONH edema may be different between the human disease and its animal models, the pathophysiological changes that ultimately result in axon ischemia are likely similar. In both clinical disease and in the nonhuman primate and rodent models, vascular decompensation and leakage results in ONH edema within a tightly restricted space. The edema in turn causes capillary compression and compromise, causing a compartment syndrome that generates axon ischemia. Axon ischemia results in subsequent localized loss of axon transport and electrical conductivity, followed by inflammation, RGC axon collapse and, eventually, isolated RGC loss. 

Acute ONH edema is present in human NAION [3], and also in the rodent and primate models [7,14]. This edema can be quantified by optical coherence tomography (OCT) in clinical NAION [15], pNAION [12], and rNAION, [14,16]. The primate retina and ONH can be evaluated using a standard OCT device such as a Heidelberg spectral domain-optical coherence tomograph (SD-OCT), and the ONH changes associated with pNAION are similar to those of clinical NAION (Figure 2A–D). However, the small size of the rodent eye requires additional optical adjustment, depending on whether or not a rodent-specialized imaging device such as a bio-optigen OCT is used. We initially developed a simple plano-concave contact lens for use in rats and mice that is now commercially available (Micro-R and Micro-M; Cantor and Nissel, UK) and that can be used with standard OCT devices such as the Heidelberg spectral domain-OCT in combination with a high-plus (28-diopter) correcting lens. The use of the plano-convex lens enables both a cross-sectional and *en face* view of the rodent retina (Figure 2E–H). 

### 2.2. ONH Vascular Leakage in the rNAION Model

Following rNAION induction, ONH vascular leakage is demonstrable within 5 h. We performed post-induction leakage analysis using indocyanine green (ICGA) and fluorescein angiography (IVFA) (Figure 3). ICG signal at the ONH was found to be more robust than fluorescein, likely because ICG is >95% protein bound, limiting diffusion [16]. We used a 5 h post-rNAION ICGA to determine if the relative degree of early leakage is useful as a biomarker for later rNAION severity (Figure 3I–G leakage). While all induced eyes showed some degree of ICG-leakage signal, the degree of early leakage by either method was not sufficiently robust to predict the ultimate level of rNAION severity, as measured by RGC loss at 30 days [16]. Animals with strong early signals can ultimately have little neuronal loss, whereas animals with less intense leakage could yield significant loss of RGCs (Figure 3). These findings may be associated with a number of factors, including ON sheath resistance, and the relative size of each animal’s ON. Thus, very early biomarkers (<1d) of rNAION lesion severity remain to be identified. 

### 2.3. ONH Edema-Based Expansion Predicts rNAION Severity

While very early biomarkers of severity remain to be found, edema development in the intraocular portion of the ONH at 1–2d post-induction using (mean) ONH diameter has proven to be a useful indicator of ultimate RGC loss [14,16]. The mean diameter of the intraocular portion of the uninduced rat ONH for current purposes, is defined as the mean distance between the two margins of the inner nuclear layers on either side of the optic nerve head, and is 310 ± 38.2 µm [16]. A 6-s laser induction generates ONH ICG leakage, edema, with ONH expansion to a mean of 347.5 ± 13.6 µm. However, this level of edema does not result in discernable RGC loss. Following a 9–11-s induction time, mean ONH diameter increased with a range between 355 µ to >650 µm [16]. RGC loss was only consistently seen when ONH diameter was >510 µm [16]. The majority of ONH edema develops by 1 day, with a minimal amount of additional edema between 1 day and 2 days post-induction. 

ONH expansion to >510 um post-induction is predictive of significant RGC loss. This suggests that, in preclinical trials of neuroprotective agents for NAION, OCT can be used to either identify and segregate animals for analysis in neuroprotection treatment studies with long time window effects (>1 d), or to identify agents that effectively suppress post-induction edema 1–2 days post-rNAION induction. The use of the 510 um cutoff is also useful for identifying individuals that can be included in neuroregeneration studies, since these treatments are typically given later (=days) post-induction.

### 2.4. RGC Quantification

Many RGC neuroprotection analyses use estimation of the number of surviving RGCs. Rats have only a single layer of RGCs, unlike primates. Thus, randomized statistically valid cell analysis (stereology) of RGCs can be performed easily on retinal flat-mount preparations by immunohistochemistry. Antibodies against Brn3a(+) identify RGC nuclei, whereas anti-RPBMS labels RGC soma. While retrograde labeling of RGC soma can be performed using either intracerebral injection of fluorogold into the superior colliculus or application of the fluorescent dye DiI onto the cut section of the ON, these latter approaches may not consistently label all RGCs or even the same region of RGCs. Previous studies have revealed that to achieve 90–95% RGC fluorogold labeling, the superior colliculus must be covered with fluorogold-soaked gelfoam (M. Vidal-Sanz, personal communication). 

### 2.5. Measuring Visual Function Loss following rNAION Induction

We currently evaluate visual function using a device which combines contrast sensitivity and optokinetic nystagmus (Optomotry). The optomotry device can evaluate visual acuity in each eye of an animal. Visual function in mice and rats is fairly coarse (mice: 0.5 cycles/degree; Long-Evans pigmented rats: 1 cycle/degree; ~20/500) [17]. This relatively low acuity is likely based on a lack of strong evolutionary pressure for high-level visual function in rodents compared with their motor skills. Visual acuity of albino rat strains is even lower (~0.5 cycles/degree) [18]. Thus, attempts to identify subtle degrees of subjective visual acuity improvement resulting from various neuroreparative therapies other than simple quantification of RGC numbers before and after treatment are likely to be problematic.

Visual function also is quantifiable using objective measures such as the visual-evoked potential (VEP), which evaluates signal received at the visual cortex, the photopic negative response (phNR), and the pattern electroretinogram (pERG) [19,20], the latter being able to evaluate the degree of RGC activity within the retina itself, rather than as an action potential to the brain. We previously demonstrated that in pNAION, the VEP and pERG were significantly correlated (r = 0.80, *p* = 0.0002 [12]. VEPs also may be useful in evaluating the degree of ON functional preservation distinct from simple RGC preservation. We found that administration of a monoclonal antibody (11c7mAb) that targeted the NOGO receptor in degenerate myelin did not improve either ONH edema or RGC counts following induction of rNAION, but flash VEP amplitudes were significantly preserved (*p* = 0.01 vs. vehicle; *n* = 8 animals) in 11C7mab-treated animals compared with mAb (anti-ragweed IgG) treated controls [21]. There was also reduced inflammatory cell infiltration. This suggests that treatments targeting functional preservation, distinct from simple RGC survival, may be an effective neuroregenerative strategy.

### 2.6. RGC Loss Variability in rNAION Using Different Induction Times

Generally, we use an 11-s laser exposure at 50mW laser power to induce rNAION. Power output is measured at the point of exposure, using a thermocouple power meter. These induction parameters result in few uninduced animals or unwanted intraretinal complications.

We evaluated the relative efficiency of laser exposure to induce rNAION and the corresponding severity of subsequent RGC loss (Figure 4). A 9-s laser induction (*n* = 7) yielded a mean RGC loss of 35.8 ± 17.9% (mean ± sem) (Figure 4A). The range varied from <5% to >76% RGC loss (Figure 4B). The severity variability of the 9- and 10-s-induced animals is strongly correlated with the degree of ONH edema produced. This suggests the rNAION model exhibits a threshold effect, in which low levels of ONH edema may not produce sufficient ONH compression to generate RGC loss. An 11-s induction time (*n* = 10) generated a mean 71.4 ± 2.4% (sem) RGC loss (Figure 4A) with a range of 60% to >84% loss (Figure 4B). Although these data support the idea of a threshold effect, longer exposure times were associated with a greater number of unwanted intraretinal complications; eg, branch retinal vein occlusions (BRVOs) and central retinal vein occlusions (CRVOs). Intraretinal complications can skew interpretation of neuroprotective approaches for rNAION, since retinal neuroprotective mechanisms (which include activity of Mueller cells) also may be considerably different from the astrocyte-driven mechanisms of the ON. This is confirmed by the number of agents known to be retina-neuroprotective but which fail to induce optic nerve neuroprotection. Therefore, identification and elimination of animals with retinal ischemic complications by both visual (retinal-fundus) examination and SD-OCT imaging is important in designing relevant treatment trials, especially those with fewer animals. As previously mentioned, ganzfeld electroretinography also can be helpful in identifying animals with large degrees of retinal ischemia. 

### 2.7. Early Ischemic Stress-Related Changes in RGCs

Following rNAION induction, RGCs with axon dysfunction begin to show significant stress. cfos elevation is detected as soon as 5 h post-induction, which was the earliest time analyzed (see Figure 7a in [5]). In an early experiment, we used Affymetrix microarray analysis to evaluate early stress-related retinal gene expression changes at 1-day and 3 days post-induction from pooled (*n* = 3 retinae) Poly (A+) RNA from each condition, with two replicates (Table 1). Complete Affymetrix chip data is shown in Appendix A.

The most dramatic changes in gene expression compared with contralateral uninduced eyes were found at 1d post-induction (Timp1: 25-fold induction; Stat1: 10-fold induction, and Lipocalin-2: 7-fold induction). STAT1 and lipocalin-2 expression declined to 4-fold and 2.4-fold, respectively, by 3 days. The decline in retinal TIMP1 expression was even greater, to 1.6-fold at 3 days post-induction, suggesting that there are distinct retinal requirements associated with early RGC stress.

Affymetrix-derived retinal results following rNAION-induced stress were compared against results obtained from changes in ischemic ONH gene expression obtained by illumina deep sequencing. Sequence data was obtained from total RNA from isolated ONHs from naïve vs. 3d post-rNAION induced animals (previously reported in [22]). Using the combined datasets, we then evaluated protein expression and RGC localization for a number of candidate RGC stress genes. These included Lipocalin-2, Jun-phosphorylation (pJun) and Corticotrophin releasing factor/releasing hormone (CRF/CRH). cJun phosphorylation is closely associated with neuronal ischemic stress [23]. CRF was chosen bibliographically because although this gene is known to be constitutively expressed in retinal amacrine neurons [24], it is upregulated in neurons after ischemia [25]. We confirmed expression in RGCs using Brn3a colocalization (Figure 5). 

RGCs whose axons are undergoing ischemic stress rapidly upregulate cJun phosphorylation, likely via retrograde signaling through the double leucine zipper kinase (DLK), a member of the JNK signaling pathway [26]. One day post-rNAION induction, nearly all RGC nuclei express some degree of pJun signal, although the signal varies in intensity (Figure 5A). In contrast, while strong lcn-2 upregulation is seen at the mRNA level in Table 1, lipocalin-2 protein signal is low (Figure 5B). However, 3 days post-induction, lcn2 expression is clearly detectable in nearly all RGCs, as defined by coexpression of green (lcn2) and Brn3a(+) (red) nuclei (Figure 5E). There is also a significant reduction in the intensity of Brn3a(+) signal at this time (Figure 5E; compare red signal with Brn3(+) signal in 5B). RGC stress is also confirmed by the strong expression of pJun at 3 days (Figure 5D; green nuclei), with few RGC nuclei still expressing strong Brn3a(+) signal (Figure 5D; red nuclei). Individual confocal channels reveal that, while most RGC nuclei still express Brn3a, few RGCs have the same signal level seen in naïve retinae, suggesting a shutdown of Brn3a-homeodomain-related gene functions by this time. The timing of these events provides an indirect estimation of the potential window of neuroprotective treatment opportunities in NAION. 

### 2.8. Soluble and Cellular Inflammatory ONH Responses

It is not surprising that a robust inflammatory response quickly follows the induction of an ischemic axonopathy. Nevertheless, the existence of post-NAION inflammation was debated for years, until the rNAION model became available. This was simply because of a lack of relevant clinical material, since NAION is not associated with acute mortality. Post-rNAION inflammatory responses include both soluble cytokine/chemokine changes in the lesion site within a day, as well as inflammatory cell migration (both extracellular macrophages and microglia) within 3 days post-lesion. 

We performed deep sequencing on both naïve rat ONH and pooled ONH tissue from 1d post-rNAION induced eyes (*n* = 6). These results are shown in Table 2, and the data is derived from the Appendix A previously deposited [22]. IL6 showed the greatest increase 1d post-induction (1.66 ± 1.0 fold in rNAION/Naïve), followed by CCL17 and CCL9 (1.46 ± 1.0 and 1.40 ± 0.85 fold, respectively).

Within the lesion itself, early release (1d) of inflammatory proteins include multiple pro-inflammatory cytokines, such as osteopontin/SPP1 (see Table 1) and interleukin-6 (IL6; see Table 2) (rNAION/naïve = 1.53 and 1.66-fold, respectively) [22]. IL6 is a cytokine with pleotropic (multiple actions) activity. It is important in induction of acute-phase inflammation-related proteins, but also in the transition to chronic inflammation [27,28]. Additional changes include inflammatory prostaglandins (PGE_2_) and thromboxanes A1 and A2. Post-rNAION, the retina and ONH also express pro-inflammatory TNFα and IL-1β mRNAs, with levels peaking by the first day and declining thereafter.

### 2.9. Neuroprotective Treatment Approaches to rNAION 

#### 2.9.1. Edema/Vascular

ONH edema is a key factor in NAION pathology. Early edema reduction would thus would seem to be an effective approach to NAION treatment. The caveats with respect to this approach are: (1) treatment needs to be effective after initiation of the edema-inducing insult, but before irreversible ischemic axon damage occurs, and (2) treatment must be effective before irreversible death-associated signaling occurs in the RGCs. 

PGJ_2_. 15-Deoxy *delta* 12,14-Prostaglandin J_2_ (PGJ_2_) is a natural, non-enzymatically generated product of prostaglandin D2 (PGD_2_). We selected this prostaglandin because early studies revealed that PGD synthase gene expression declined dramatically after rNAION induction and then rebounded to supernormal levels precisely when post-stroke edema resolved, suggesting that the loss of either PGD_2_ or one of its metabolites might play a major role in edema development and its rebound have a role in edema resolution.

PGJ_2_ administered in pharmacological doses (100 ug/kg) up to 5 h post-induction reduced ONH edema (see Figure 1 in [29]) and also reduced RGC loss in rats with rNAION [29], and in a model of old-world primates [30]. Ultrastructural studies revealed that PGJ_2_ likely reduces early intraneural edema by inhibiting endothelial vesicular fluid transport after rNAION induction [29]. This probably occurs by PGJ_2_’s suppression of NFκB activation in disc vasculature [31]. Late (>1d) treatment with PGJ_2_ alone did not notably affect the levels of late ONH edema and did not improve RGC survival. This suggests that PGJ_2_’s major neuroprotective effect in rNAION occurs by early edema suppression.

2.TRPM4/Sur-1 inhibition. Following CNS ischemia, astrocytes express the transient receptor potential cation-family M/sulfonylurea receptor-1 (TRPM4/SUR1). This results in opening of a water channel, with resulting intracellular astrocytic edema contributing to ischemic damage. SUR1 inhibitors such as glibenclamide have been shown to be neuroprotective after hemorrhagic and ischemic strokes as well as after spinal cord injury in humans [32,33,34]. Interestingly, SUR-1 is not expressed in early (<1d) ONH exhibiting edema after rNAION induction [35], and there is minimal elevation in ONH-SUR1 mRNA expression even 3d post-induction compared with naïve ONH (DS-Seq: naïve 9.34 vs. 3d post-induction: 9.85). We performed a limited sighting study (*n* = 4 animals), utilizing rNAION induction in both eyes of each animal, with each animal serving as its own induction control. Animals were induced in one eye (11 s/50 mW/1 mg/kg 2.5 mM rose Bengal). Immediately following induction, we administered glibenclamide (50 ug/kg) daily via intraperitoneal injection for the next 5 days. One week post-induction of the first eye, the contralateral eye was induced to an identical level, and vehicle (PBS) was administered intraperitoneally. ONH edema was evaluated using SD-OCT at 2d post-induction in each eye (Figure 6A). We also evaluated RGC survival in both eyes of all animals. Because of the small number of animals used, no statistically valid analysis was possible.

Glibenclamide treatment resulted in only a small reduction in ONH edema, compared with the vehicle-treated animals (Figure 6A). RGC quantification 30d post-induction revealed essentially no difference in RGC survival in glibenclamide-treated vs. vehicle-treated animals (775.7 ± 149.1 glibenclamide vs. 767.6 ± 167.6 cells sem for vehicle; Figure 6B). Interestingly, astrocytic SUR-1 induction takes about 1 day post-injury in the CNS (Gerzanich, personal communication), suggesting that TRPM4/SUR-1 suppression is not likely to be neuroprotective in sudden ON edema-associated damage. Nevertheless, other non-SUR1-dependent astrocytic mechanisms responsible for ON edema may still be responsive to edema reductive therapies.

#### 2.9.2. Other rNAION-Neuroprotective Mechanisms

Astrocyte-associated neuroprotective mechanisms. Astrocytes exert neuroprotective effects directly via both edema-related mechanisms [36,37,38] and mechanisms related to ischemic preconditioning and that are associated with A1-Adenosine agonist activity [39]. In addition to TRPM4/SUR-1-mediated channels, Adenosine receptor-1 and -3 (A1R, A1e) agonists have been shown to be neuroprotective in cerebral ischemia [40] and are expressed in A2B5(+) glia, which include astrocytes [41]. A1R are also strongly expressed in RGCs [42,43]. Activation of A1Rs is at least partially responsible for neuroprotection associated with ischemic preconditioning [44]. Deep sequencing of rat ONH reveals moderately high levels of the A1R and nearly 8-fold less of A3Rs (Ds-Seq analysis in naïve ONH 12.53 (A1R) vs. 3.84 (A3R)(data provided on mdpi website https://www.mdpi.com/article/10.3390/cells10061440/s1, accessed on 5 June 2022 Appendix A). This suggests that A1 or A1R agonists might be useful in RGC neuroprotection. We evaluated topical administration of Trabodenoson, a highly selective, topical A1R agonist. Although Trabodenoson only moderately suppressed ONH edema (564 ± 14.52 um sem for vehicle vs. 512.15 ±20.12 um sem for Trabodenoson; *n* = 13), the difference was significant (two-tailed *t*-test: *p* = 0.05). Even more importantly, however, Trabodenoson was potently RGC neuroprotective, with RGC loss reduced from 53.2 ± 7.1% sem in vehicle-treated animals to 27.9 ± 6.5% sem RGC loss in animals treated with Trabodenoson (Figure 7B; data derived from [45]). These results also were statistically significant (two-tailed *t*-test; *p* = 0.01) and suggest that adenosine receptor agonism is potently neuroprotective, with effects only partly exerted through edema reduction. It should be noted that Trabodenoson treatment in this study was begun 3d before rNAION induction. Thus, the ability of A1 agonists to exert a neuroprotective effect in post-ischemic events is still an open question. We are now evaluating the ability of other A1 agonists to exert neuroprotective effects post-rNAION.

Following rNAION, cellular inflammatory invasion is present by 3 days post-induction [46], with late increases in a number of chemokines and inflammatory factors. Our lab and others have also evaluated compounds that can suppress or modify cellular inflammation, using multiple inflammatory pathways.

2.Corticosteroids. An initial clinical report suggested that immune suppression in clinical NAION treatment could be beneficial using high-dose/chronic glucocorticoids [47], but this was based on a retrospective study. In current clinical practice, some neuroophthalmologists either give no treatment or, based on the previous report, utilize corticosteroids (with its associated side effects, including steroid dependent glaucoma, diabetes and hip fractures). Most later prospective studies using steroids in clinical NAION did not show visual improvement effects [48,49]. These studies included patients diagnosed within a month of symptom onset. One study utilizing a modified steroid early in treatment suggested some effect [50], but the time from onset of symptoms to beginning treatment was not discussed. Multiple investigators have explored the use of inflammatory suppression to reduce rNAION-associated damage [51], with mixed results. Glucocorticoids were reported to be neuroprotective in a single report involving the rodent NAION model [52], but we have not seen this effect in our own study nor in other reported studies utilizing both methylprednisolone and triamcinolone [53]. No studies have combined corticosteroid administration and other inflammation- or edema-related therapies, such as the tetracyclines (see paragraph below) or PGJ_2_. This is a reasonable area for further research, and the rNAION model would appear to be ideal for this.3.Tetracycline derivatives. Minocycline is a CNS-penetrating tetracycline derivative with CNS neuroprotective effects in some models [54], but not in others, such as closed head trauma [55]. For ON injury, minocycline was reported to preserve RGCs at 2 weeks post-ON transection and at 4 weeks in model of glaucoma [56]. Minocycline treatment begun 3 days prior to rNAION was also reported to be RGC neuroprotective [57]. However, we did not see any neuroprotective effect when minocycline was begun after rNAION induction [58]. This suggests that treatments designed to inhibit inflammation need to be started early in the course of the process.

#### 2.9.3. Potentiated and Synergistic Treatment Approaches

As previously noted, no currently available treatment has been shown to be consistently clinically effective in human NAION. This may be because: (1) Multiple damage mechanisms (soma, axon, inflammation, post-stroke demyelination) are activated. (2) Late recognition of clinical NAION typically limits treatment opportunities to neuroregeneration as the window of survival for many RGC subtypes and their axons may have passed by the time NAION is diagnosed. However, recent studies have demonstrated that approaches using multiple reparative and/or regenerative mechanisms can exert a potentiated (one drug does not elicit a response on its own but enhances the response to another drug) or synergistic effect (the combined effect of two drugs is greater than the sum of their separated effects). Thus, combinatorial treatments for ON repair can potentiate ON regeneration effectiveness [59]. The rNAION model enables effective evaluation of multiple factors as well as multiple approaches; ie, combinatorial approaches with anti-inflammatory + growth factors, cell death suppression agents, etc.

We have evaluated a number of dual-treatment approaches, including the multispecific tyrosine kinase inhibitor sunitinib (Sutent), which was reported to enhance RGC survival in ex vivo culture as well as in vivo models of ON injury [60]. However, combining Sutent 10 mg/kg with PGJ_2_ did not enhance overall RGC survival compared with PGJ_2_ alone in our rNAION model, suggesting that the neuroprotective effects of suppressing nonspecific receptor tyrosine kinase activity and NFkB activity (anti-inflammatory) may mechanistically overlap. A recent approach of inhibiting **all** prostaglandin synthesis while simultaneously administering one anti-inflammatory prostaglandin (PGJ_2_), was unsuccessful due to incomplete prostaglandin suppression by a monoacylglycerol lipase (MAGL) inhibitor [22]. These results reveal some of the complexities of multiple treatment approaches, as well some of the potential drawbacks.

## 3. Discussion

The rNAION model system has enabled recognition of a vast number of variables that are potentially associated with clinical ischemic optic neuropathies such as NAION. Although simply inhibiting ONH edema can be effective in early stages of lesion development, this is not likely to prove effective in clinical practice. An awareness of the significant differences in rNAION-associated RGC survival between pre- and post-treatment therapies is also essential; pre-treatments for NAION are not likely to be effective unless they are being considered as long-term adjuncts to prevent NAION from occurring in the contralateral eye of at-risk patients or in situations where there is high risk of precipitating NAION, such as cardiac surgery with cardiopulmonary bypass.

The rNAION model can be useful to help decipher the individual contributions to pathophysiology made by vascular leakage, as well as by the individual ONH cellular components such as astrocyte-associated inflammation and neuroprotection, RGC somatic and axon response times and stress resistance mechanisms, and potential neural progenitor cell contributions to ischemic stress resistance [61]. There is also a great opportunity to identify early biomarkers to both confirm the presence of disease and to predict disease severity. 

As previously noted, no currently available treatment has been shown to be consistently clinically effective in human NAION. This is likely because: (1) Multiple damage mechanisms (soma, axon, inflammation, post-stroke demyelination) are activated and (2) The recognition of NAION typically limits neuroprotection treatment opportunities to inflammatory suppression or neuroregeneration as the window of survival for many RGC subtypes and their axons may have passed by the time NAION is diagnosed. However, recent studies have demonstrated that approaches using multiple reparative and/or regenerative mechanisms can exert a potentiated (one drug does not elicit a response on its own but enhances the response to another drug) or synergistic effect (the combined effect of two drugs is greater than the sum of their separated effects). Thus, combinatorial treatments for ON repair can potentiate ON regeneration effectiveness [60]. The rNAION model enables effective evaluation of multiple factors as well as multiple approaches; ie, combinatorial approaches with anti-inflammatory + growth factors, cell death suppression agents, etc.

The rNAION model may also be valuable for determining the potential time windows for treatment opportunities and identifying mechanisms that may extend these windows, both in terms of RGC survival, axonal survival and inflammatory pathways. Using the rNAION model to evaluate strategies for enhancing RGC and axonal survival by modulating multiple independent pathways is an exciting possibility, which may potentially extend the effective treatment time windows. The rNAION model’s mechanistic similarity with the primate NAION model also provides an invaluable bridge to understanding primate responses in ischemic optic neuropathy, and ultimately the rapid evaluation of potential therapeutics and strategies for treatment of the human disease.

## 4. Materials and Methods

While the data in the majority of this report is based on other research we have previously reported on (Section 2.1, Section 2.2, Section 2.3, Section 2.4, Section 2.5, Section 2.6, Section 2.8 and Section 2.9), the data reported in Figure 4 and Table 1 in Section 2.7 and Figure 6 in Section 2.9 have never been reported before. Thus, we include the materials and methods for these experiments.

Animals: All animal protocols were approved by the UMB institutional animal care and use committee (IACUC). 

Affymetrix gene expression analysis: Three Male Sprague Dawley rats (225–250 g) were used for each pooled mRNA preparation. Following unilateral rNAION induction, retinae were pooled from 3 individual eyes treated similarly, and the contralateral uninduced eyes were used as controls. Tissue was obtained from animals following induction at 1 and 3 days. Total RNA was purified using Qiaprep RNAeasy micro kit, and poly(A)RNA further isolated using oligo d(T) cellulose columns. 500 ng of total RNA was used for each pooled cDNA probe. Complementary DNA (cDNA) was generated from pooled poly (A+) RNA using a genechip one cycle cDNA synthesis kit for synthesis of double stranded cDNA with a T7 RNA polymerase promotor. Biotinylated complementary RNA (cRNA) probes were then generated for reaction to the genechips using a one- or two cycle in vitro transcription (IVT) reaction kit.

Two RNA preps were used for analysis of each timepoint and contralateral controls, with two U34A rat genome genechips (total of 4 chips). cRNA was loaded onto an Affymetrix U34A total genechip array, containing >24,000 genes and EST clusters, with ~7000 known genes (rat gene expression). The chips were reacted/hybridized with the biotinylated probes and processed on an Affymetrix fluidics station 400 platform according to manufacturer’s protocols, and following genechip scanning, the resulting data was analyzed using Genechip sequence analysis software at the DC Children’s hospital core sequencing facility. These results are seen in Table 1 results in Section 2.7.

Immunohistochemical analysis of rNAION-associated early RGC stress. To confirm early RGC stress following rNAION, we induced rNAION (11 s/50 mW/500 uM/532 nm) spot size in one eye each of three Long-Evans (L-E) male rats. L-E pigmented rats are preferred over albino strains for visual function studies and have largely replaced our use of the SD strain. Animals were euthanized at 1, 3 and 7 days post-induction, and tissue post-fixed in 4% paraformaldehyde-phosphate-buffered saline pH 7.4 (PF-PBS). Whole retinae were isolated, fixed overnight in PF-PBS, and the inner limiting membranes were permeabilized by brief (−70 °C) freezing in PBS, followed by incubation in 10% normal donkey serum/0.3% Triton X-100 with primary antibodies. Tissues were treated with primary antbodies. These included rabbit monoclonal antibody to lipocalin 2 (Lcn2)-purchased from Abcam (ab41105). Polyclonal goat antibody to Brn 3a (Santa Cruz Chemicals, Santa Cruz, CA). Rabbit monoclonal antibody to phospho-Jun (pJun; Cat # 3270/D47G9; Cell Signaling Technologies, Danvers, MA). Rabbit Polyclonal antibody to CRF/CRH (CRF; Thermofisher/Bioss (BS-0328R)). Tissues were washed and incubated with fluorescent-labeled secondary donkey antibodies (Jackson Immunoresearch, West Grove PA), and examined using confocal microscopy on a Leica 4 channel confocal microscope. 

Glibenclamide treatment for ONH edema. 4 male Long-Evans (LE) rats (225–250 g) were used as a sighting study to determine whether ONH edema is suppressible using glibenclamide, an FDA approved anti-glycosemic drug which, even at low doses, inhibits opening of astrocytic sulfonylurea receptor-1 (SUR1)-water associated channel [34], and suppresses cerebral edema. Animals were induced with 11 s laser (50 mW/532 nm/500 um spot size) 30 s post-injection with rose Bengal (2.5 mM solution’ 1 ml/kg). Both eyes utilized in this sighting study; Animals were treated with subcutaneous (SC) saline for 5 days post-induction of the first eye. Following induction of the second eye, animals were SC injected with glibenclamide (50 ug/kg) immediately after induction and daily for five days thereafter. We evaluated ONH edema using OCT at two days post-induction of each eye [16], and quantified RGCs by stereology using Brn3a(+) immunopositivity of flat mounted retinae, at 30d post-induction of the second eye, as previously reported [14]. These results are seen in results Section 2.9.

## 5. Patents

The use of PGJ_2_ in the treatment of NAION is covered by US patent US 8,106,096 B2 ‘Compositions and methods for treatment of optic nerve diseases (SLB).

## Figures and Tables

**Figure 1 ijms-23-15604-f001:**
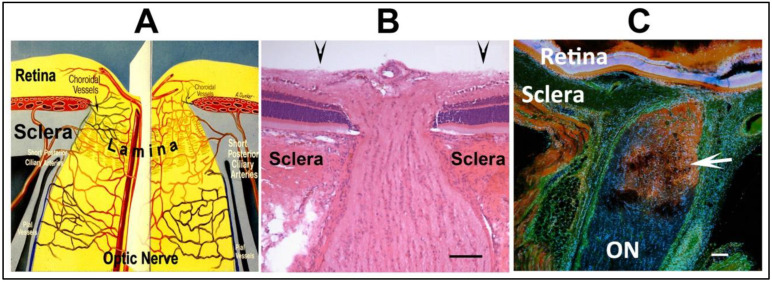
Overview of the ONH and contributory vascular components, and limitations of vascular leakage in rNAION. (**A**). Schematic of the vascular supply of the primate ONH, The ONH is partially bounded by the thick scleral wall, which resists lateral expansion. The fenestrated lamina within the ONH enables transit of the RGC axons. ONH circulation is a plexus formed with contributions from intraretinal, choroidal, optic nerve intraneural and pial circulations, as well as direct contribution from the short posterior ciliary arteries (SPCAs) in primates. (**B**). H&E-stained cross-section of the rat ONH. The unmyelinated portion of the ONH is bounded by the sclera, and smaller in diameter than the myelinated distal ON. The arrowheads above the figure indicate 500 µm boundary. (**C**). Limits of ONH vascular leakage following rNAION induction (arrow) 1d post-induction in the rNAION model. Immunohistochemistry using labeled 3 kDa (red-Dextran) and 55 kDa (green-FITC/BSA) probes. Serum leaks from decompensated capillaries in the ONH in a focal manner, with distal ON vasculature unimpaired. Scale bar: 100 µm in (**B**,**C**).

**Figure 2 ijms-23-15604-f002:**
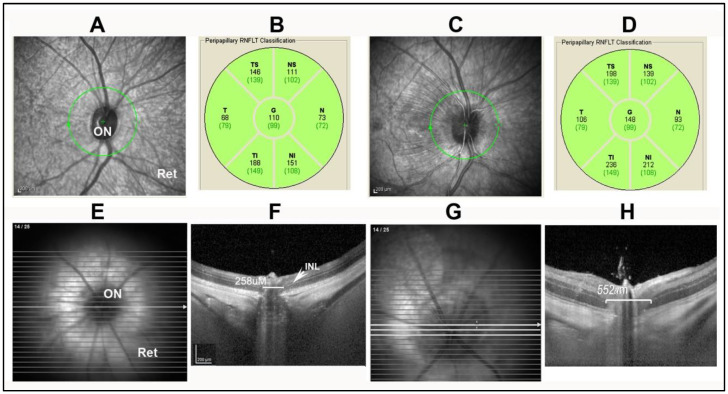
Comparison of OCT appearance of the ON in two NAION (pNAION and rNAION) models: (**A**). OCT image of uninduced NHP. (**B**). Quantification of peripapillary nerve fiber layer (RGC axonal layer) thickness in uninduced NHP. The global nerve fiber layer thickness (**G**) is 110 µm, with quantification of individual regions. (**C**). OCT image of 2d post-induced NHP ONH. There is edema of the ONH, with striae extending out into the superior retina. (**D**). Quantification of the RGC axon layer. There is increased thickness of the global peripapillary nerve fiber layer, indicating ONH edema. (**E**). OCT *en face* view of uninduced rat ON. The ONH is dark, and relatively small in diameter. (**F**). Cross section enabled by planoconcave contact lens. The width of the ONH at mid-diameter, as measured by the distance in microns between the boundary of inner nuclear layer (INL) on either side of the nerve, is 258 µm. (**G**). OCT *en face* view of rNAION 2d post-induced ON. The nerve head is pale and enlarged. (**H**). ONH cross sectional view after rNAION induction. The ONH diameter (in brackets) has expanded to 552 µm; compare with panel F, indicating ONH edema.

**Figure 3 ijms-23-15604-f003:**
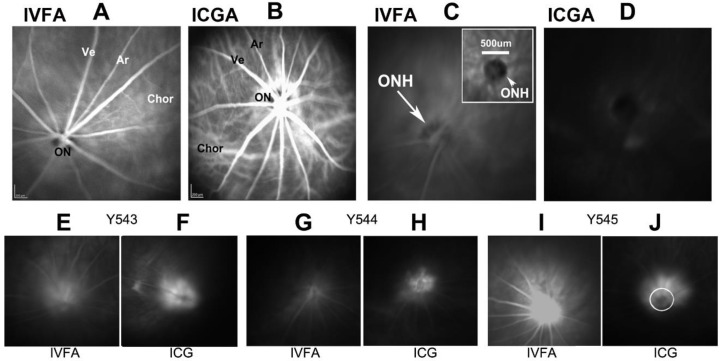
Markers of ONH vascular leakage in rNAION. (**A**,**B**). Baseline retinal fundus photos taken 30 s post-intravenous dye injection in a naïve (uninduced) animal of A. Fluorescein (IVFA). The intraretinal veins (Ve) and arteries (Ar) are visible, but the optic nerve circulation is muted due to the presence of melanin pigment, which blocks visible light. (**B**). Indocyanine green (ICGA). ICG fluorescence is visualized via infrared light, which passes through melanin. The choroidal circulation below the retinal pigment epithelium is clearly visible, as is the enhanced circulation around the ON. (**C**). fluorescein dye fluorescence 5 min post-injection. The ONH is barely visible. (**D**). ICG fluorescence 5 min post-injection. No leakage is visible. (**E**–**J**): fluorescein and ICG image pairs of the same eyes in three animals (Y543, Y544, Y545) taken 5 h post-rNAION induction. There are significant differences in the amount of post-injection fluorescence revealed by the two dye types. In general, the ICG dye signal is more stable, due to its nearly 100% binding to serum proteins and reduced diffusion. The ICG patterns are indicative of the deeper leakage in the ONH. The circle in plate (**J**) is 500 µm. Data for Figure 3 is reprinted with permission from [16]. 2021 PLOS-One.

**Figure 4 ijms-23-15604-f004:**
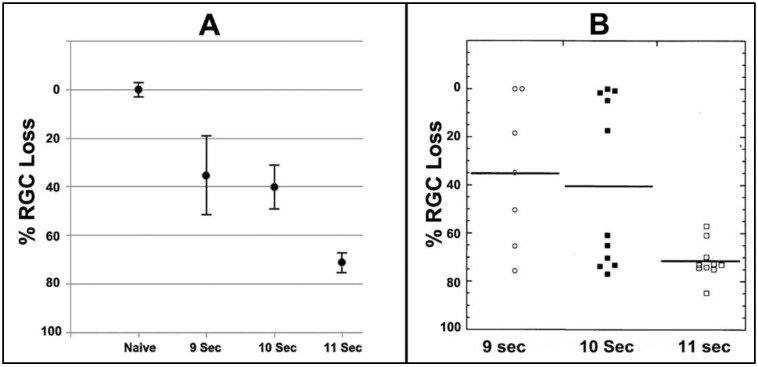
RGC loss patterns with increasing amounts of laser exposure. (**A**). Comparison of mean RGC loss in rNAION with increasing exposure times using a 532nm laser at constant power and spot size (50 mW power; 500 um spot size). (**B**). Induction times compared against the range of individual animal RGC loss values. Low induction times (9 s) result in a wide range of RGC loss values, including a number of animals without any RGC loss as well as animals with minimal levels of RGC loss. An intermediate induction time (10 s) results in identification of two animal subgroups suggestive of a threshold effect, with subthreshold- (low levels of RGC loss) and threshold- animals (high-level RGC loss). A further increase in exposure time to 11 s results in the majority of animals having significant RGC loss (threshold). Data for Figure 4 is reprinted with permission from [16]. 2021 PLOS-One.

**Figure 5 ijms-23-15604-f005:**
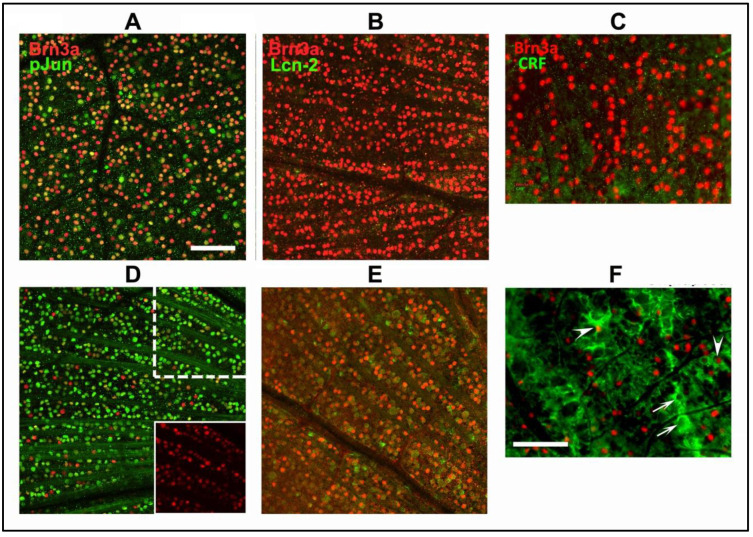
RGC stress protein expression after axon ischemic stress. A-C: RGC responses 1d post-rNAION induction. D-E: RGC responses 3d post-rNAION induction. F. RGC responses 7d post-rNAION induction. (**A**,**D**): Phospho-Jun (pJun). The majority of RGC nuclei (Brn3a(+) nuclei; in red) colocalize with pJun (in green) within 1 day of induction. (**D**). pJun expression is demonstrable in nearly every RGC nucleus in an affected region by 3d post-induction. The upper (dashed line-outlined actual area) and lower (inset: solid line) indicated areas reveal the relative decrease in Brn3(+) expression in RGCs by 3d post-induction. (**B**,**E**): Lipocalin-2. There is little signal at 1 day, despite high levels of lipocalin-2 mRNA. Lipocalin-2 protein signal is present in the majority of affected RGCs by 3d post-induction. (**C**,**F**): CRF. CRF expression is minimal in RGCs at one day (**C**), while in (**F**) Strong expression is seen 7 days post-induction. Expression is prominent in cells that have lost their nuclear Brn3a expression (arrows), but also in nuclei with persistent Brn3a (arrowheads). Scalebars in (**A**,**F**): 50 uM.

**Figure 6 ijms-23-15604-f006:**
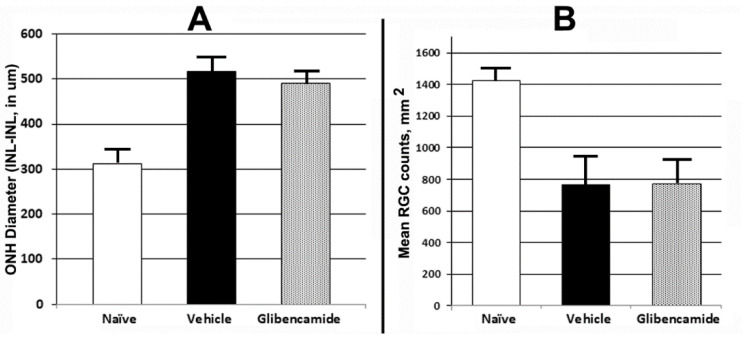
Effects of an astrocytic TRPM4/SUR1 (white matter edema inhibitor) on ONH edema and RGC survival post-rNAION. Adult rats were administered glibenclamide (40 ug/kg) immediately after rNAION induction and daily for 4 subsequent days via intraperitoneal injection. (**A**). ONH edema 2d post-induction. Minimal reduction was seen in glibenclamide-treated animals (ONH mean values 516 ± 17µm in vehicle-treated vs. 492 ± 20µm s.e.m in glibenclamide-treated animals; *n* = 4). (**B**). RGC survival post-rNAION induction. Brn3a(+)-RGC counts in naïve retinae yielded a ratio of 1423.2 ± 19.3 cells/unit area (*n* = 10 animals). Retinae 30d post-induction yielded a mean of 767.6 ± 167.6 cells/unit area for vehicle treatment vs. 775.7 ± 149.1 cells/unit area for glibenclamide treatment.

**Figure 7 ijms-23-15604-f007:**
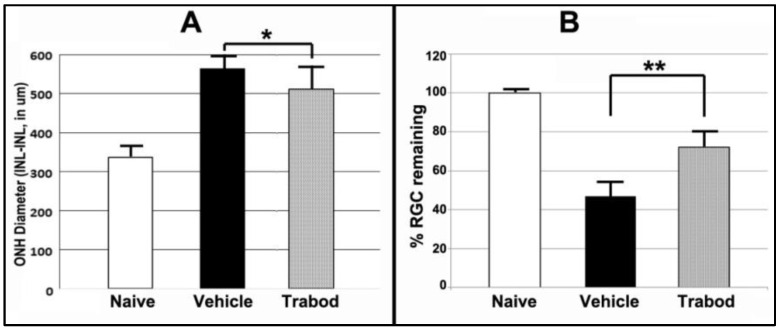
Adenosine A1 agonist effects on ONH edema and RGC survival. Trabodenoson drops were administered twice daily for 3 days prior to rNAION induction and for 2 weeks post-induction. (**A**). ONH edema. There is a statistically significant 15.4% reduction in ONH edema in Trabodenoson-treated eyes compared with vehicle-treated eyes. Two-tailed *t* test: * *p* = 0.049. (**B**). RGC preservation. Trabodenoson treatment resulted in a 62.1 ± 6.5% (sem) RGC survival 30d post-rNAION induction compared with 46.8 ± 7.1 % (sem) RGC survival in vehicle-treated eyes. Two-tailed *t* test: ** *p* = 0.015.

**Table 1 ijms-23-15604-t001:** Changes in retinal and ONH gene expression post-rNAION induction. Retinal expression data are derived from Affymetrix U34A genechips (Data from Appendix A, this report), whereas the ONH (in red) gene expression data are derived from Illumina sequencing and previously reported in Appendix A in [22]. Retinal ratios are generated from individual assay results between comparison sets. ONH ratio data are generated from a single sample of six pooled rNAION ONHs from 1d post-induction, divided by the mean results from five individual naïve ONHs. Genes evaluated for protein expression (seen in Figure 4) are indicated in **bold**. Lipocalin-2 mRNA expression was elevated >6 fold at 1 day and decreased to >2 fold by 3 days post-rNAION. Crh expression did not increase in the first 3 days.

Gene ID	Gene Symbol	Gene Name	1d Post-rNAION Ratio (R1/L1) (Retina)	3d Post-rNAION Ratio (R3/L3) (Retina)	1d Post-rNAION Ratio (R3/L3) (ONH)
M14656	Spp1/osteopontin	Secreted phosphoprotein 1	3.432111001	1.417575157	1.530076555
AA892553	Stat 1	STAT1	10.43670886	4.104424779	1.080385971
U66707	Densin-180/Lrrc7	Densin-180	1.713261649	2.555331992	1.093128466
AF030089	Ania-4/Dclk1	neurotransmitter early gene 4 (ania-4)	8.987133516	1.700414108	0.975284552
J02722	hmox1	Heme Oxygenase-1	3.646239554	1.928571429	1.132401488
L33869	Cp	Ceruloplasmin	2.937233461	2.498855835	1.000494963
AF008650	SLC-1/SSTR1	Somatostatin receptor-1	3.253018661	2.729357798	0.954323595
D28508	JAK3	Jak-3 protein kinase	8.614443084	1.719766472	1.142099095
AI045030	CELF	CELF protein	4.181425486	1.86818377	1.027053789
M19651	Fra-1 fosl1	Fos-related antigen	5.223048327	1.319474836	1.174091099
AI176456	MT-2a	Metallothionein 2a	6.331651318	1.597822606	0.971216928
M29866	C3	Complement 3c	4.054368932	3.743093923	0.939352616
AI169327	TIMP1	Tissue inhibitor-metalloproteinases-1	25.57614679	1.648916117	1.223379175
U53184	Litaf	Lipoprotein inducing TNF factor	2.894752945	1.301528426	1.016124153
**AA946503**	**Lcn2**	**Lipocalin-2**	**6.955862534**	**2.376604473**	**1.038533352**
AA893280	Plin2	Perilipin 2	6.457792208	2.39125239	1.104502556
**M54987**	**CRF/Crh**	**Corticotropin releasing hormone**	**0.923984891**	**1.05511811**	**0.933884278**
U92081	RTI40/PDPN	RNRTI1-podoplanin	4.873994638	1.848832659	0.999564836
U56407	Ube2d4	ubiquitin conjugating enzyme	4.523255814	4.270042194	0.915009744
M11597	Calca	alpha-type calcitonin gene-related peptide	4.830188679	2.294117647	1.559015745
X61381	IFITM3	Interferon induced mRNA	3.532416331	0.916807529	1.048079385
U09228	TCF4	Deaconess E-box binding factor	3.228754366	1.075950771	1.041957282
D00753	Cpi-26	contrapsin-like prot. inhibitor (CPi-26)	4.660706861	2.730576441	Not Found
X06769	cFos	cFos mRNA	2.259195894	0.880295098	1.076489824
AF082125	AHR	Aryl hydrocarbon receptor	4.016318205	1.222446237	1.077670043
U01344	NAT1	N-Acetyl Transferase	3.98401066	1.263825929	1.025181547
X17053	MCP-1/CCL2	Monocyte chemoattractive protein-1	3.333156499	1.466160658	1.12708558
**X17163**	**cJun**	**c-jun oncogene**	**1.38262668**	**1.140383102**	**1.043633554**

**Table 2 ijms-23-15604-t002:** ONH-Interleukin and chemokine responses in early (1d) rNAION), compared with naïve tissue. Mean naïve is the average of five individual samples (Naïve 1–5), whereas rNAION is a pooled sample containing ONH-RNA from six eyes 1d post-induction. IL6 showed the greatest overall increase at 1 day.

Gene Symbol	Gene Name	Naive_1	Naive_2	Naive_3	Naïve_4	Naive_5	Mean Naïve	rNAION	rNAION /Naïve
	**Interleukins**								
Il6	interleukin 6	3.66	4.85	3.66	4.37	6.01	4.51	7.50	**1.66**
Il27	interleukin 27	4.77	5.05	3.66	4.93	3.66	4.41	5.72	**1.30**
Il24	interleukin 24	3.66	3.66	3.66	3.66	3.66	3.66	4.70	**1.28**
Il10	interleukin 10	3.66	3.66	3.66	3.66	3.66	3.66	4.49	**1.23**
Il3	interleukin 3	3.66	3.66	3.66	3.66	3.66	3.66	4.49	**1.23**
Il1b	interleukin 1 beta	6.35	5.67	5.79	5.62	7.05	6.10	7.43	**1.22**
Il11	interleukin 11	5.52	5.73	7.38	5.09	10.46	6.84	8.26	**1.21**
	**Chemokines**								
Ccl17	C-C motif chemokine ligand 17	3.66	3.66	3.66	4.79	6.00	4.35	6.34	**1.46**
Ccl9	chemokine (C-C motif) ligand 9	3.66	4.51	3.66	4.69	5.72	4.45	6.25	**1.40**
Cxcr3	C-X-C motif chemokine receptor 3	3.66	5.62	3.66	5.37	3.66	4.39	5.95	**1.35**
Cxcl11	C-X-C motif chemokine ligand 11	4.55	5.25	3.66	6.44	6.62	5.31	7.10	**1.34**
Cxcl3	chemokine (C-X-C motif) ligand 3	3.66	3.66	3.66	3.66	6.10	4.15	5.50	**1.33**
Ccl22	C-C motif chemokine ligand 22	4.92	4.51	3.66	3.66	6.03	4.56	5.95	**1.31**
Ccr7	C-C motif chemokine receptor 7	5.71	4.51	5.76	4.99	4.70	5.13	6.67	**1.30**
Cxcl10	C-X-C motif chemokine ligand 10	6.91	6.74	7.41	7.34	8.75	7.43	9.62	**1.29**
Ccl3	C-C motif chemokine ligand 3	6.10	5.48	3.66	5.73	7.46	5.69	7.36	**1.29**
Ccl4	C-C motif chemokine ligand 4	5.12	5.19	3.66	5.21	7.04	5.24	6.76	**1.29**
Ccl11	C-C motif chemokine ligand 11	5.98	6.00	5.24	6.27	8.80	6.46	8.03	**1.24**
Ccr4	C-C motif chemokine receptor 4	3.66	3.66	3.66	4.37	3.66	3.80	4.70	**1.23**
Ccl21	C-C motif chemokine ligand 21	3.66	3.66	3.66	3.66	5.98	4.13	5.06	**1.23**
Ccr8	C-C motif chemokine receptor 8	3.66	3.66	3.66	3.66	3.66	3.66	4.49	**1.23**
Cxcl17	C-X-C motif chemokine ligand 17	6.20	5.81	3.66	5.84	5.92	5.48	6.66	**1.21**
Cxcl6	C-X-C motif chemokine ligand 6	5.26	4.96	6.17	5.40	6.61	5.68	6.81	**1.20**

## Data Availability

Data from Table 1 can be obtained from Appendix A in this paper. Data from Table 2 can be obtained from Appendix A previously deposited in figshare: https://doi.org/10.3390/cells10061440, and previously published in [22]. Accessed on 6 June 2022.

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
