# Peer review of "Neuroprotection and Neuroregeneration Strategies Using the rNAION Model: Theory, Histology, Problems, Results and Analytical Approaches"

_ijms, 2022, doi:10.3390/ijms232415604_

Round 1

Reviewer 1 Report

The manuscript entitled Neuroprotection and Neuroregeneration strategies using the rNAION model: theory, histology, problems, results and analytical approaches focuses on a very important and interesting topic regarding NAION modeling. However, there are some major issues that need to be addressed by the authors.

Comments regarding English language and style

In general, English language should be improved with emphasize on style (for example there are numerous sentences with same word (terms) repeated, often sentences are not clear and are difficult to read). I suggest an overall improvement and rewriting the manuscript.

Examples of minor English errors:

Line 41-42: “Part of the reason for this lack of progress in clinical treatment has been a lack of understanding of the pathophysiological mechanisms” – please rephrase the sentence, the word lack is repeated twice.

Line 50: …”and analysis of time windows of opportunity for treating this disorder”- please rephrase

Line 69: “Scale bar: 100µ in B and C” – did the authors refer to 100µm? Please correct.

Line 70-71: “Because the ONH is bounded within the thick-walled sclera and because in primates, the individual axon bundles are enclosed between collagenous columns, there is little room for compensatory expansion in the setting of edema.”

Line 102-104: Unlike the collagen sheaths in the human ON that are quite dense, the sheaths surrounding rodent ON axon fascicles are extremely thin, and the laser/rose Bengal-induced vascular decompensation thus results in focal serum leakage in the ONH (Fig 104 1C), resulting in intraneural edema limited by the sclera and ON sheath (Fig 1C).

These are just few examples, there are numerous issues such as abovementioned trough the whole manuscript.

Materials and methods

1.      Why did the authors performed Affymetrix gene expression analysis on Male Sprague-Dawley rats and Immunohistochemical analysis of rNAION-associated early RGC stress on Long-Evans male rats? Why two different strains?

2.       The nature of this article is rather unclear…Is this a research article reporting novel authors’ results or a review article? If this is not the review, then the fact that most of this report is based on other research previously published (Sections 1-6, 8, and sections 9A1, 9A2 and 9B, as stated in Materials and Methods) is unacceptable. On the other hand, if this article represents a review, then it should be written in a different style and in much more concise and clearer manner.

General remarks

Although this article reports numerous and valuable results regarding rNAION model and, in general, human NAION, it lacks strong and clear aim. In addition, it is written in a vary unclear manner, there is no structure and is difficult to follow. I wholeheartedly suggest addressing these issues in order to improve the manuscript.

Author Response

We appreciate the thoughtful review done by the reviewer, and attach our comments. We explain what we did and the change made in italics.  Reviewer 1 had significant requests for changes; These are addressed first and below: 

Reviewer 1

The manuscript entitled Neuroprotection and Neuroregeneration strategies using the rNAION model: theory, histology, problems, results and analytical approaches focuses on a very important and interesting topic regarding NAION modeling. However, there are some major issues that need to be addressed by the authors.

Comments regarding English language and style

  1. In general, English language should be improved with emphasize on style (for example there are numerous sentences with same word (terms) repeated, often sentences are not clear and are difficult to read). I suggest an overall improvement and rewriting the manuscript.

We have reviewed the entire manuscript for clarity and removed repeated phrases and word when necessary. We have also completely reviewed the manuscript and improved the sentences for clarity. This can be seen in the ‘Final with tracked changes’ copy.

Overall nature of the article

  1. The nature of this article is rather unclear…Is this a research article reporting novel authors’ results or a review article? If this is not the review, then the fact that most of this report is based on other research previously published (Sections 1-6, 8, and sections 9A1, 9A2 and 9B, as stated in Materials and Methods) is unacceptable. On the other hand, if this article represents a review, then it should be written in a different style and in much more concise and clearer manner.

We agree with the reviewer that the combination of research data in an otherwise review article is confusing, since there is a mixture of both review and new data. To clarify, this manuscript was initially requested (and written) as an invited review, but the opportunity to explain specific aspects of the model’s mechanism with new data was considered so useful and illuminating to the field that we included data that had been held back (since 2003 with the Affymetrix retinal stress data; Table 1), that were confirmed by new immunohistochemical studies on retinal ganglion cell stress (Figure 5). We also included a sighting study utilizing glibencamide (Figure 6), which confirms the functional conclusions we drew in a previously published study (Nicholson et al, ref 38). It was these valuable inclusions that required rewriting and resubmission as a research article with the corresponding methods. We feel that these inclusions greatly strengthens and enhance the manuscript. If this is not acceptable, the new data will be removed and ultimately submitted as separate report(s), and the paper again re-submitted as a review article. We would prefer to keep this article in the present form.

Examples of minor English errors:

  1. Line 41-42: “Part of the reason for this lack of progress in clinical treatment has been a lack of understanding of the pathophysiological mechanisms” – please rephrase the sentence, the word lack is repeated twice.

Rewritten and first ‘lack’ is removed. Changed to: One reason for this lack of progress in clinical treatment is a dearth of knowledge of the pathophysiological mechanisms responsible for permanent visual loss after NAION.

  1. Line 50: …”and analysis of time windows of opportunity for treating this disorder”- please rephrase

Changed to: ‘and to identify the time windows of opportunity for treating this disorder[5-7].  ‘

  1. Line 69: “Scale bar: 100µ in B and C” – did the authors refer to 100µm? Please correct.

Corrected

  1. Line 70-71: “Because the ONH is bounded within the thick-walled sclera and because in primates, the individual axon bundles are enclosed between collagenous columns, there is little room for compensatory expansion in the setting of edema.”

Rewritten

  1. Line 102-104: Unlike the collagen sheaths in the human ON that are quite dense, the sheaths surrounding rodent ON axon fascicles are extremely thin, and the laser/rose Bengal-induced vascular decompensation thus results in focal serum leakage in the ONH (Fig 104 1C), resulting in intraneural edema limited by the sclera and ON sheath (Fig 1C).

Corrected. The revised statement reads” While the collagen sheaths in the human ON  surrounding the axon fascicles are quite dense, the sheaths surrounding rodent ON axons are extremely thin. The laser/rose Bengal-induced vascular decompensation causes focal serum leakage in the ONH (Fig 1C), resulting in intraneural edema limited by the sclera and ON sheath (Fig 1C).”

These are just few examples, there are numerous issues such as above mentioned trough the whole manuscript.

 We have revised the entire MS to make it easier to follow.

Materials and methods

  1. Why did the authors performed Affymetrix gene expression analysis on Male Sprague-Dawley rats and Immunohistochemical analysis of rNAION-associated early RGC stress on Long-Evans male rats? Why two different strains?

We state:

“L-E pigmented rats are preferred over albino strains for visual function studies and have largely replaced our use of the SD strain”

Reviewer 2 Report

This manuscript is perfect and completed. I would like to accept it with this form.    

Author Response

Thank you for your kind review. We appreciate your efforts and understanding.

Reviewer 3 Report

Overall a well thought out article with substantial justifications on recommendations that is worth publishing.

I see that you have mentioned in that no current effective treatment that is consistently effective. You have addressed Hayreh et al. in line 523 regarding corticosteroids, and quoted 2 articles with small sample sizes that demonstrated them not working. In current clinical practice, most neuro-ophthalmologists do nothing, while some base their practice from Hayreh's work. It is either long term steroids (and associated side effects) or nothing.

Hayreh has also responded to the quoted articles about the study designs (timeframe and modality of determining vision improvement) and described how they cannot determine that steroids do not work.  As I understand from your article, the other treatment modalities are only tested in animal models. Is there consideration of combining corticosteroids with your proposed methods (eg PGJ, tetracycline)? Is there more to discuss regarding Hayreh's work (eg. recognising immediate reduction of edema can possibly confer permanent visual field loss), or even using his work to bridge human and animal studies of treating NAION?

Line 168: Define NHP

Was there a reason for "materials and methods" to follow the "discussion" section?

Author Response

We thank the reviewer for his comments and criticism.

We have included the reviewers suggestions about clinical use of steroids in NAION, and also that combinatorial therapies utilizing both corticosteroids and either tetracyclines or PGJ2 might be a reasonable approach to test in the rNAION model. We have rewritten the paragraph:

‘2.        Corticosteroids. An initial clinical report suggested that immune suppression in clinical NAION treatment could be beneficial using high-dose/chronic glucocorticoids [50], but this was based on a retrospective study. In current clinical practice, some neuroophthalmologists either give no treatment or, based on the previous report, utilize corticosteroids (with its associated side effects, including steroid dependent glaucoma, diabetes and hip fractures). Most later prospective studies using steroids in clinical NAION did not show visual improvement effects [51,52]. These studies included patients diagnosed within a month of symptom onset [52]. One study utilizing a modified steroid early in treatment suggested some effect [53], but the time from onset of symp-toms to beginning treatment was not discussed. Multiple investigators have explored the use of inflammatory suppression to reduce rNAION-associated damage [49], with mixed results. Glucocorticoids were reported to be neuroprotective in a single report involving the rodent NAION model [54], but we have not seen this effect in our own study nor in other reported studies utilizing both methylprednisolone and triamcinolone [55]. No studies have combined corticosteroid administration and other inflammation- or edema-related therapies, such as the tetracyclines (see paragraph below) or PGJ2. This is a reasonable area for further research, and the rNAION model would appear to be ideal for this.’

Line 168: Define NHP

We define this on page 4:

….as occurs in humans and in the nonhuman primate (NHP) NAION model (pNAION) [7,13].

Was there a reason for "materials and methods" to follow the "discussion" section?

We incorporated the materials and methods section at the end as suggested by the IJMS template.

Round 2

Reviewer 1 Report

Dear Authors,

The manuscript still seems confusing and there is no overall improvement. I would strongly recommend the suggested corrections in terms of rewriting the manuscript to fit into either research or review article. 

Kind regards